# Highly efficient frequency conversion with bandwidth compression of quantum light

Markus Allgaier[1], Vahid Ansari[1], Linda Sansoni[1], Christof Eigner[1], Viktor Quiring[1], Raimund Ricken[1], Georg Harder[1], Benjamin Brecht[1,2] & Christine Silberhorn[1]

Hybrid quantum networks rely on efficient interfacing of dissimilar quantum nodes, as elements based on parametric downconversion sources, quantum dots, colour centres or atoms are fundamentally different in their frequencies and bandwidths. Although pulse manipulation has been demonstrated in very different systems, to date no interface exists that provides both an efficient bandwidth compression and a substantial frequency translation at the same time. Here we demonstrate an engineered sum-frequency-conversion process in lithium niobate that achieves both goals. We convert pure photons at telecom wavelengths to the visible range while compressing the bandwidth by a factor of 7.47 under preservation of non-classical photon-number statistics. We achieve internal conversion efficiencies of 61.5%, significantly outperforming spectral filtering for bandwidth compression. Our system thus makes the connection between previously incompatible quantum systems as a step towards usable quantum networks.

[1] Integrated Quantum Optics, Applied Physics, University of Paderborn, Paderborn 33098, Germany. [2] Clarendon Laboratory, Department of Physics, University of Oxford, Oxford OX1 3PU, UK. Correspondence and requests for materials should be addressed to M.A. (email: markus.allgaier@uni-paderborn.de).

Photons play the important role of transmitting quantum information between nodes in a quantum network[1]. However, systems employed for different tasks such as generation, storage and manipulation of quantum states are in general spectrally incompatible. Therefore, interfaces to adapt the central frequency and bandwidth of the photons are crucial[2–4]. To achieve any viable bandwidth compression, the interface has to provide at least a net gain over using spectral filters. Electro-optical frequency conversion can provide such high efficiencies for bandwidth compression[4] and shearing[5] of quantum pulses. However, it is limited to frequency shifts of a few hundred gigahertz. Optical frequency conversion in nonlinear crystals offers both large frequency shifts as well as high conversion efficiencies[6–10]. Operating on chirped pulses allows to perform spectral shaping[11], an approach with which a bandwidth compression of 40 has been demonstrated[2,12], however, with low efficiencies below spectral filtering. Reaching high conversion efficiencies with this method is challenging, as very broad phasematching is required, which in turn limits the allowed interaction lengths and hence the conversion efficiencies. An alternative approach is to engineer the phasematching of the sum-frequency process itself[13] by choosing appropriate group velocity and pump-pulse conditions. Such engineering has been widely exploited for parametric downconversion (PDC)[14–17] to produce decorrelated photon pairs efficiently. For frequency conversion, this approach has not been investigated.

The quantum pulse gate (QPG)[9,18–20] is such a device that exploits specific group-velocity conditions: The input and the pump are group-velocity matched, while the output is strongly group-velocity mismatched. This is achieved in a type-II sum-frequency process in a periodically poled titanium-indiffused waveguide in lithium niobate. The group-velocity matching ensures that spectrally broad input pulses overlap throughout the crystal while the mismatch with the output in combination with the long interaction length inside the waveguide results in a narrow output spectrum. Furthermore, the output temporal mode, that is, the temporal or spectral amplitude of the output pulse, only depends on the phasematching and not on the pump or input fields. This allows to convert any input to the same narrow output. It can thus interface broad PDC sources as well as narrower and even dissimilar emitters, such as quantum dots.

To demonstrate the performance of the QPG as an interface, we focus on its application as a link between PDC sources and quantum memories to produce on-demand single photons. Ideally for quantum networks, single photons are generated into well-defined optical modes and feature compatibility with low-loss fibre networks. Heralded photons from engineered, single-pass PDC fulfill these requirements[21,22]. One class of quantum memories, Raman quantum memories, can exhibit very broad spectral bandwidths of a few gigahertz[23] up to 20 GHz (refs 24,25). Long storage times have been achieved in alkali vapour memories with bandwidths of up to several gigahertz[26]; however, these are narrowband compared with the above-mentioned PDC sources with bandwidths in the terahertz regime[22]. In diamond, terahertz bandwidth can be achieved[27], but both storage time and memory efficiency are low, such that these memories cannot be utilized in quantum networks, yet. In principle, schemes exist to match both systems directly by using a very broad memory[27] or strong spectral filtering of a correlated PDC source[28], but these come at the expense of short storage times or reduced purities through spectral filtering[29]. A bandwidth-compressing interface between the broadband PDC sources at telecom wavelengths and the narrower quantum memories at visible or near-infrared wavelengths is therefore desirable.

We show in this work that dispersion engineering can be used to develop processes that provide spectral reshaping and high conversion efficiencies at the same time. We demonstrate such an interface by converting single photons from 1,545 nm and a bandwidth of 1 THz to 550 nm and a bandwidth of 129 GHz under preservation of the second-order correlation function $g^{(2)}(0)$ while achieving external conversion efficiencies high enough to outperform a spectral filter producing an equivalent output spectrum.

## Results

**Experimental setup and spectral properties of the PDC source.** Our experimental setup is depicted in Fig. 1. We generate single photons at 1,545 nm from an 8 mm long type-II PDC source in periodically poled potassium titanyl phosphate with a poling period of 117 μm and a Klyshko efficiency[30] of 20.2%. The pump beam for the PDC source is created by a series of elements, starting with a Ti:Sapphire mode-locked laser, which pumps an optical parametric oscillator, followed by second-harmonic generation and bandwidth fine-tuning with a 4*f* spectral filter. The bandwidth is adjusted to ensure a decorrelated PDC state. We characterize the spectral properties of the PDC photons by measuring their joint spectral intensity with a time-of-flight spectrometer[31], consisting of a pair of dispersive fibres and a low-jitter superconducting nanowire single-photon detector (SNSPD; Photon Spot). From this measurement, shown in Fig. 2a, we conclude that the bandwidth (full-width at half-maximum) of the idler photon is $963 \pm 11$ GHz at 1,545 nm central wavelength. Furthermore, the round shape of the joint spectral intensity and the Schmidt number[32] $K = 1/\sum_k \lambda_k^2 = 1.05$ extracted from the measurement indicate that the photon pairs are indeed spectrally decorrelated. $\lambda_k$ are the weights of the Schmidt modes. We keep the pulse energy of the PDC pump at a low level of 62.5 pJ to ensure that mainly photon pairs and only few higher-photon-number components are created. At the output, an 80 nm wide band-pass filter centred at 1,550 nm is used to filter out background processes while not cutting the spectrum of the actual PDC process.

The heralded idler photon is then sent to the QPG, consisting of a periodically poled 27 mm long LiNbO₃ crystal with Ti-indiffused waveguides and a poling period of 4.4 μm. It is pumped at 854 nm with light from the same Ti:Sapphire laser, which is spectrally shaped by means of another 4*f* line containing a liquid-crystal spatial-light modulator. The modulator can be used to adapt the QPG pump to any input. To characterize the

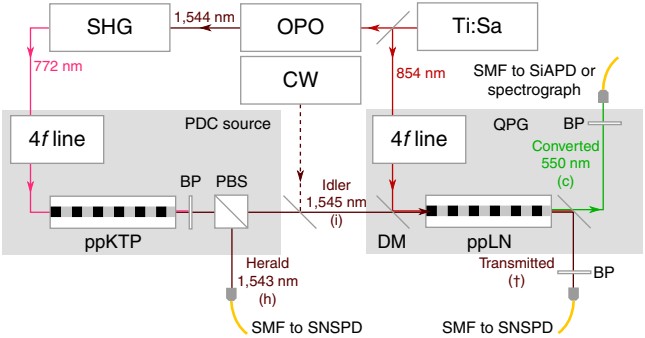

**Figure 1 | Experimental setup.** Setup used for characterization of the transfer function of the the quantum pulse gate (QPG) as well as the measurement of conversion efficiency, correlation functions and spectra. BP, band pass filter; CW, continuous wave laser; DM, dichroic mirror; OPO, optical parametric oscillator; PBS, polarizing beam splitter; ppKTP, periodically poled potassium titanyl phosphate crystal; ppLN, periodically poled lithium niobate crystal; SHG, second-harmonic generation; SMF, single mode fibre; Ti:Sa, Ti:Sapphire laser.

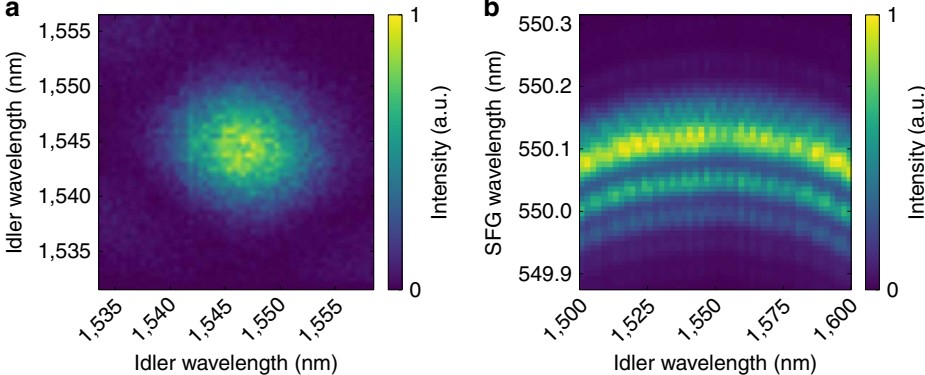

**Figure 2 | Spectral characteristics of the parametric downconversion and sum-frequency generation.** (**a**) Joint spectral intensity of the photon pairs generated in the PDC source. The spectra were measured using two dispersive fibre time-of-flight spectrometers. (**b**) Phasematching function of the quantum pulse gate. The spectrum of the sum-frequency generation (SFG) signal from the Ti:Sapphire laser and a tunable continuous wave telecom laser were recorded on a Czerny–Turner spectrometer.

QPG, we measure its phasematching function by recording the sum-frequency signal of a broad pump and a tunable continuous wave telecom laser on a Czerny–Turner spectrograph equipped with 2,398 lines mm$^{-1}$ grating and a single-photon-sensitive electron multiplying charge-coupled device camera. The result is shown in Fig. 2b. The horizontal orientation of the phasematching function is due to the fact that the input and pump are group-velocity matched, while the output is strongly group-velocity mismatched. This leads to the narrow spectrum of the output field while accepting a broad input field. As the slope of the phasematching function is connected to the group-velocity mismatch between input and pump, the horizontal portion on the top indicates perfect group-velocity matching, where the output spectrum depends only on the phasematching and not on the pump. This holds for a telecom input bandwidth as large as 20 nm. As the PDC photons are only 7.8 nm wide, we are well within that range and adjust the pump bandwidth accordingly to ensure maximum conversion efficiency. After the conversion, we separate both the converted and the unconverted light from the background and residual pump by means of broadband filters and couple all fields into single-mode fibres. It is noteworthy that the phasematching bandwidth and therefore the bandwidth compression depend on the sample length and could therefore be increased or decreased to get the desired output. As the group-velocity curves steepen towards shorter wavelength, moving the process in this direction would increase the group-velocity mismatch between input and output resulting in even greater bandwidth compression.

**Noise properties of the conversion**. To be viable as an interface in quantum networks, the device has to leave the quantum nature of the single photons untouched. To measure this, we employ photon-number statistics, namely, the heralded second-order autocorrelation function of the photons measured with a 50/50 beam splitter and two click detectors:

$$g^{(2)} = \frac{P_{cc}}{P_1 \cdot P_2},\tag{1}$$

where $P_{cc}$ is the coincidence probability and $P_1$ and $P_2$ the single-click probabilities. The QPG does not change the $g^{(2)}(0)$, which takes the value of $0.32 \pm 0.01$ both before and after the frequency conversion. With $g^{(2)}(0) < 1$, the single-photon character is verified before and after the conversion. The value before the conversion can be explained with higher photon number components. More notably, there are no measurable noise

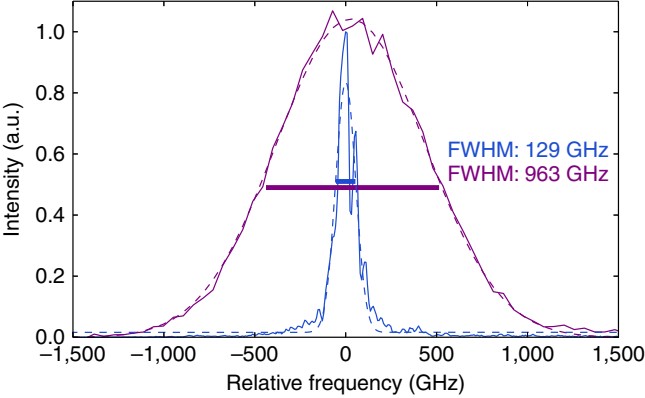

**Figure 3 | Marginal spectra before and after conversion.** Marginal spectra of the PDC idler photon before (magenta) and after (blue) frequency conversion in the quantum pulse gate centred around their respective centre frequencies. Dashed lines correspond to Gaussian fits from which the bandwidths were obtained. The spectrum of the idler photons were measured using a dispersive-fibre time-of-flight spectrometer. A Czerny–Turner spectrograph was used for the spectrum of the converted light.

photons added polluting the $g^{(2)}$ in the frequency conversion process.

**Bandwidth compression and efficiency**. To estimate the bandwidth compression, we record the spectrum of converted PDC photons with the aforementioned Czerny–Turner spectrometer. The marginal spectra of the idler photon together with the converted spectrum are depicted in Fig. 3. The converted light has a spectral bandwidth of $129 \pm 4$ GHz and a central wavelength of 550 nm. Compared with the original bandwidth of 963 GHz of the PDC photon, this implies a bandwidth-compression factor of $7.47 \pm 0.01$.

The second, equally important figure of merit is the conversion efficiency. If the conversion efficiency is low, a simple spectral filter could outperform the device. Were the idler converted by a continuous wave pump, the bandwidth would remain constant at 963 GHz. Filtering down to 129 GHz would then imply a throughput of $13.40 \pm 0.02\%$ (the error corresponds to the fit errors for the spectral bandwidths in Fig. 3), assuming the conversion itself is lossless. To measure the conversion efficiency, we send the photons to SNSPDs and a silicon avalanche

photodiode (SiAPD) for infrared or visible photons, respectively. We estimate the internal efficiency of the process itself as well as the external efficiency including all optical loss in the setup. As a measure for the internal efficiency, we use the depletion of the transmitted light by calculating the Klyshko efficiency[30] $\eta_t$ of the unconverted 1,545 nm light, transmitted through the QPG with the QPG pump open and blocked. The Klyshko efficiency is defined as $\eta_t = P_{cc}/P_h$, where $P_{cc}$ is the coincidence-count probability between the herald (h) and unconverted, transmitted (t) PDC photon (refer to labels in Fig. 1) and $P_h$ is the herald-count probability alone. From this depletion, we get the internal conversion efficiency of the process

$$\eta_{int} = 1 - \frac{\eta_t^{open}}{\eta_t^{blocked}} \qquad (2)$$

where the superscript denotes whether the QPG pump was blocked, meaning that the idler mode is merely coupled and transmitted through the QPG, or open and the conversion process takes place. Using the depletion of the unconverted light has the advantage that it provides a direct measure of the internal conversion efficiency. By contrast, one would need precise knowledge of all losses to estimate it from the upconverted signal. The resulting value for the internal conversion efficiency is 61.5%.

As a measure for the external conversion efficiency, we use the ratio between the Klyshko efficiencies of the converted light $\eta_c$ and the unconverted idler light before the QPG $\eta_i$, corrected only for the different detection efficiencies of the SiAPD compared with the SNSPD:

$$\eta_{ext} = \frac{\eta_c \cdot \eta_{SNSPD}}{\eta_i \cdot \eta_{SiAPD}} \qquad (3)$$

where $\eta_{SNSPD} = 0.9$ and $\eta_{SiAPD} = 0.6$ are the detector efficiencies of the SNSPD and SiAPD photon detectors, respectively. This external conversion efficiency is 16.9%. Owing to some spatial mode mismatch, the coupling of the converted light into a single mode fibre is reduced compared with the unconverted light. Taking into account this reduced fibre compatibility of the green mode (50% instead of 80% for the herald), the external efficiency amounts to 27.1%. This can be seen as the free-space efficiency of the device. As all of these efficiencies result from counting sufficiently large numbers of photons, errors are negligible. The coupling of the green mode into the fibre can be further improved by optimizing the waveguide structure or the coupling optics. The

difference between the internal and the external efficiency is mainly due to linear optical losses in uncoated lenses and a a 4f line band-pass filter, with a total transmission of 68% and a waveguide-incoupling efficiency of around 71%.

These conversion efficiencies show that the QPG offers useful bandwidth compression and provides a net gain over using a spectral filter. For the first time, this is realized in combination with substantial frequency conversion. This is true not only when looking at the internal conversion efficiency but even when comparing to the external conversion efficiency, which already includes all losses, such as waveguide and even fibre couplings.

## Discussion

Having demonstrated a viable interface, we calculate the process parameters required to interface the proposed broadband memories in diamond based on nitrogen and silicon vacancy centres[24,25]. The degrees of freedom available for tuning the conversion process are primarily the temperature and the choice of the nonlinear material. As a basis for this study, we use effective Sellmeier equations[33,34] of the modes inside the waveguide. Figure 4a shows the group-velocity mismatch between PDC idler and pump at two different temperatures. The two light stripes in the colour code represent areas with zero group-velocity mismatch for 190 °C (left stripe) and 300 °C (right stripe), whereas the solid white lines indicate wavelength combination where the sum-frequency is at the desired output frequency. The main target wavelength in this work, the transition of a charge-neutral nitrogen vacancy centre (NV0) in diamond[24] at 574 nm, can be addressed with a group-velocity-matched sum-frequency generation process at a sample temperature of 300 °C. The PDC wavelength would be at 1,560 nm and the pump at 907 nm, well within reach of PDC sources and Ti:Sapphire laser systems. For the proof-of-principle experiment in this work, we have chosen a slightly different operating point of 190 °C as it simplifies the choice of suitable ovens and insulation materials, thus shifting the target wavelength to 550 nm. As unwanted effects such as photorefraction are only present at lower temperatures[35], there is no fundamental limitation for increasing the temperature as high as the Curie temperature. The alternative silicon-vacancy transition[25] at 738 nm cannot be reached with the birefringent properties of lithium niobate. However, lithium tantalate, a less birefringent material, supports it. The signal wavelength of this process could be at 1,278 nm and the pump wavelength at

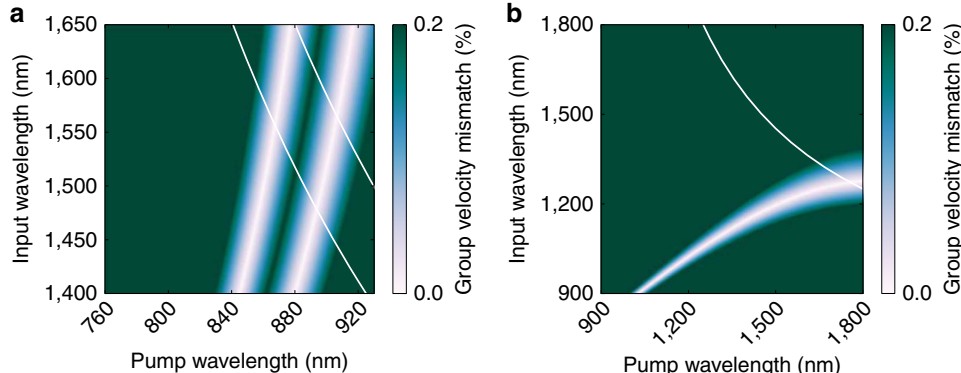

**Figure 4 | Group-velocity mismatch in lithium niobate and lithium tantalate. (a)** Group-velocity matching in Ti:LiNbO₃ between waveguide modes in ordinary and extraordinary polarization at two different temperatures (left stripe: 190 °C, right stripe: 300 °C). The solid white lines indicate wavelength combinations where the sum-frequency generation process reaches the desired wavelength of 574 nm (right line) for the transition of the charge neutral nitrogen vacancy centre or the wavelength of 550 nm (left line) chosen in this article. **(b)** Group-velocity matching in bulk LiTaO₃ between the ordinary and extraordinary polarization at 190 °C. Here the white line indicates an output wavelength of 738 nm, corresponding to the silicon vacancy transition in diamond.

**Table 1 | Herald and coincidence count rates before and after the quantum pulse gate (QPG) used to obtain the external conversion efficiency.**

|  | Before QPG | After QPG |
|---|---|---|
| Herald counts ($s^{-1}$) | 430,000 | 465,000 |
| Coincidence counts ($s^{-1}$) | 86,000 | 10,600 |
| Klyshko efficiency | 20.2% | 2.27% |

**Table 2 | Count and coincidence rates for measuring the second-order correlation function before and after the quantum pulse gate (QPG).**

|  | Before QPG | After QPG |
|---|---|---|
| Herald counts ($s^{-1}$) | 910,000 | 970,000 |
| Coincidences herald—mode1 ($s^{-1}$) | 6,900 | 3,900 |
| Coincidences herald—mode2 ($s^{-1}$) | 7,200 | 2,780 |
| Triple coincidences ($s^{-1}$) | 18.0 | 3.42 |

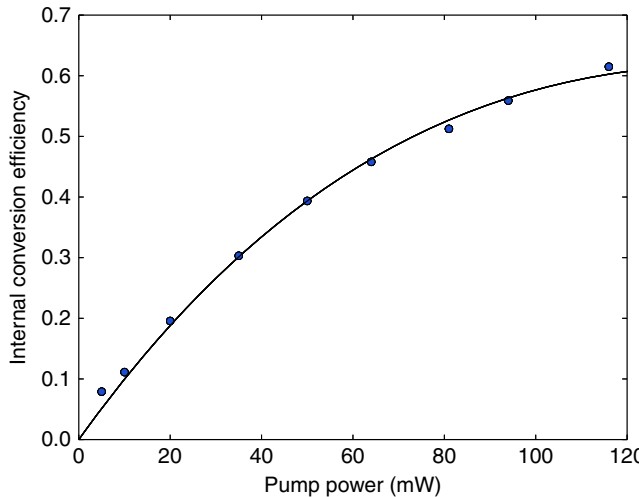

**Figure 5 | Pump power dependence of the internal conversion efficiency.** Pump power dependence of the quantum pulse gate's internal conversion efficiency. The solid line was fitted to the data and follows $0.619 \cdot \sin^2(0.130 \cdot \sqrt{P})$. Each data point was obtained by measuring the depletion of the unconverted (transmitted) beam's count rate. Poissonian distributed statistical uncertainties of the count rates are small, error bars are therefore omitted as they are smaller than the data points.

1,748 nm or vice versa. Temperature tuning of the group-velocity matching in the same way as in lithium niobate can also be considered. Figure 4b shows the parameter space for that process. Note that these numbers are based on bulk Sellmeier equations[36] and might slightly differ for waveguides. Apart from sum-frequency processes, difference-frequency generation can also be considered. For example, conversion of near-infrared light as emitted by semiconductor quantum dots to the telecom band can be carried out with an infrared pump such as the one employed in ref. 37. Overall, a large range of wavelengths can be covered with the available materials and realistic process parameters.

In conclusion, we have realized a device that not only offers efficient upconversion from telecom light to the visible spectrum but also useful bandwidth compression. As the phasematching bandwidth is proportional to the inverse of the sample length, the compression factor is in principle scalable. It is noteworthy that the device does not provide a fixed bandwidth ratio between input and output but rather a fixed output bandwidth, such that the same converter can be used for inputs of different bandwidth.

## Methods

**Laser system.** The main laser system employed in the experiment is a Coherent Chameleon Ultra II Titanium Sapphire laser with an APE Compact OPO optical parametric oscillator. The pulse duration of the Ti:Sapphire oscillator is 150 fs at a repetition rate of 80 MHz. The optical parametric oscillator's pulse duration is 190 fs. Its emission at 1,545 nm is converted to 772.5 nm by a periodically poled bulk lithium niobate second-harmonic generation crystal fabricated in-house in Paderborn. The bandwidth of the 772.5 nm light used as the PDC pump is 3 nm (all bandwidths given as full-width at half-maximum). A Photonetics Tunics continuous wave laser was used in the characterization of the QPG's phasematching.

**Spectral pump shaping.** Two $4f$ line pulse-shaping setups are employed in the experiment. Both use a dispersive element to separate spectral components. The spectrum is then manipulated in the focal plane of a lens. The one for the PDC pump is a folded geometry prism monochromator with an adjustable slit as described in ref. 22. The resolution is 0.7 nm. The pump for the QPG can be intensity and phase-shaped with a liquid-crystal-on-silicon-based spatial light modulator setup in a folded grating monochromator geometry with a resolution of 22 pm. In this work, the PDC pump spectrometers was set to the full 3 nm bandwidth to match the phasematching bandwidth of the PDC crystal in order to achieve a decorrelated PDC state. The QPG pump was set to 6 nm.

**Photon pair source.** The PDC photon pair source is a commercially available periodically poled potassium titanyl phosphate crystal with rubidium-exchanged waveguides purchased from ADVR. The crystal is 8 mm long with a poling period of 117 µm over a poled length of 6 mm. The source is pumped to produce a

decorrelated photon pair state with a bandwidth of 7.8 nm. The Schmidt number obtained from the measured joint spectral intensity is 1.05, and the Klyshko efficiency is 20.2%. A coincidence window of 5 ns was used to obtain this number.

**Time-of-flight spectrometer.** The time-of-flight spectrometer consists of two dispersive fibres introducing group delays of 431 ps nm$^{-1}$ each. The chirped photons are then detected by superconducting nanowire single-photon detectors manufactured by Photon Spot combined with a AIT TTM8000 time tagger. The convoluted jitter in the coincidence measurement is 150 ps leading to a spectral resolution of 0.35 nm.

**Coincidence measurements.** The coincidence window for all coincidence measurements was set to 5 ns. For the measurement of external conversion efficiencies, we measured Klyshko efficiencies, that is coincidence rates devided by herald counts. Table 1 shows the herald and coincidence count rates leading to the external efficiency discussed in the paper.

The measurements were conducted over periods of 30 s before the QPG and 46 s after, yielding 12 million and 21 million herald counts, respectively. The herald is not sent through the pulse gate; the fluctuation is due to fluctuations of the laser output power. For the $g^{(2)}$ measurement, the PDC photons in mode 1 were split up by a fibre beam splitter and all counts were conditioned on a click in the herald arm. The result was normalized over the herald counts:

$$g^{(2)} = \frac{P_{hab}}{P_{ha} \cdot P_{hb}} \cdot P_h \tag{4}$$

where a and b label the two modes resulting from splitting up mode 1. The count and coincidence rates in this measurement are shown in Table 2:

The measurement durations were 57 s in front and 190 s behind the QPG, yielding a total of 52 million and 185 million herald counts, respectively. The duration of the measurement for the converted light was increased in order to yield the same statistical error for the $g^{(2)}(0)$.

**Pump power dependence of conversion efficiency.** Figure 5 shows the dependence of the QPG conversion efficiency on the pump power. Although the conversion probability follows a $\sin^2$ dependence, we cannot reach unit efficiency with the pump power available to us.

**Code availability.** The code used to generate the findings of this study is available from the corresponding author on reasonable request.

**Data availability.** The data that support the findings of this study are available from the corresponding author on reasonable request.

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

## Acknowledgements

This work was funded by the Deutsche Forschungsgemeinschaft via SFB TRR 142 and via the Gottfried Wilhelm Leibniz-Preis.

## Author contributions

M.A. and V.A. carried out the experiment. M.A. wrote the manuscript with support from G.H. C.E., V.Q. and R.R. fabricated the $LiNbO_3$ sample. L.S., G.H. and B.B. helped supervise the project. B.B. and C.S. conceived the original idea. C.S. supervised the project.

## Additional information

**Competing financial interests:** The authors declare no competing financial interests.

**Publisher's note**: 

