## [Peer Review File · Nature Communications]

Reviewers' comments:

Reviewer #1 (Remarks to the Author):

This article by Allgaier and co-workers reports an experimental demonstration of frequency translation and bandwidth compression of a heralded single photon through a nonlinear optical frequency up-conversion. The authors employed dispersion engineering to demonstrate the bandwidth compression. High conversion efficiency and non-classical statistics of converted light field have been clearly shown. The subject is certainly of interest for researchers within the field of quantum optics and nonlinear optics and to the best of my knowledge this is indeed the first reported experimental demonstration with dispersion engineering. However, I find that the details of the experiment are rather unclear, and some efforts of clarification should be made by the authors before I recommend the paper to be published anywhere.

Major Comments

1) In this article, the authors should provide at least the following experimental conditions: the pulse duration and/or bandwidth of the light sources, the length and poling period of non-linear crystals, the observed efficiencies used in eq. (2) and (3), and the coincidence time window and measured coincidence counts.

2) The authors obtained $g(2)(0)=0.32$ in the experiment. However, they did not provide the origin of the degradation of $g(2)(0)$ in the article. The noise properties of the nonlinear optical frequency conversion are an important issue for the application to quantum network. Thus this should be discussed in the article.

3) In addition, pump power dependency of the conversion efficiency and the amount of noise should be provided because they are also important for the quantum properties of the converted light.

Minor Comments:

1) The definitions of the following variables should be clearly described.

η^{open}_t and η^{blocked}_t in eq. (2), η_{SNSPD} and η_{SiAPD} in eq. (3)

2) 4-f spectral filter should be clearly described.

Reviewer #2 (Remarks to the Author):

The manuscript describes both frequency conversion and bandwidth compression of quantum light by a nonlinear process in a Titanium-diffused Lithium Niobate waveguide. Frequency and bandwidth manipulation are vital tools in the construction of a quantum network as they can link various components with disparate wavelength requirements.

The conversion is achieved with an overall efficiency greater than what could be attained with spectral filtering, which is ascertained by group velocity matching and choosing an appropriate pump bandwidth. The authors verify that the second-order correlation function of the output light from the conversion is well below the classical value of 1, and measure a spectrum of the output confirming bandwidth compression. The authors then discuss how to adapt their setup for use with a diamond NV memory.

The results are clear, novel, and of immediate interest to the quantum optics community. The paper is well-written and accessible to a broad audience. I recommend publication in Nature Communications after the authors address the following issues:

1. The paper is lacking experimental detail. Most notably, there is no mention of photon detection

rates. What is the coincident photon rate coming out of the PDC source? After the QPG? How many photon counts were required for the g_2 measurement?

Other missing experimental details include:

- a description of the 4-f spectral filter for the source pump
- the resolution of the time-of-flight spectrometer
- coincidence detection window used in the experiment

2. The authors report that photon pairs from the source are “indeed spectrally decorrelated”. A numerical measure of separability of the initial JSI should be given to support this. Reporting the Schmidt number would suffice.

3. When discussing bandwidth compression by chirped pulse upconversion, reference should also be given to PRA 91, 033809 (2015), which discusses how high efficiency can be achieved with this method.

Answer to reviewer #1:

This article by Allgaier and co-workers reports an experimental demonstration of frequency translation and bandwidth compression of a heralded single photon through a nonlinear optical frequency up-conversion. The authors employed dispersion engineering to demonstrate the bandwidth compression. High conversion efficiency and non-classical statistics of converted light field have been clearly shown. The subject is certainly of interest for researchers within the field of quantum optics and nonlinear optics and to the best of my knowledge this is indeed the first reported experimental demonstration with dispersion engineering. However, I find that the details of the experiment are rather unclear, and some efforts of clarification should be made by the authors before I recommend the paper to be published anywhere.

We thank the reviewer for the comments which help us to improve the manuscript. We are happy to provide further details on the experiment. We added key numbers to the main text, in addition we provide a methods section in the end of the paper going into detail regarding his/her comments.

Major Comments

1) In this article, the authors should provide at least the following experimental conditions: the pulse duration and/or bandwidth of the light sources, the length and poling period of non-linear crystals, the observed efficiencies used in eq. (2) and (3), and the coincidence time window and measured coincidence counts.

The methods section on page 5 contains an overview over the laser systems employed in the experiment including the used bandwidths for pumping the non-linear processes. The parameters on the photon pair source are also given in the methods section on page 6, the QPG's poling period was added in the main text on page 3. A section on the coincidence measurements including all photon counts and heralding efficiencies can be found under methods on page 6.

2) The authors obtained $g(2)(0)=0.32$ in the experiment. However, they did not provide the origin of the degradation of $g(2)(0)$ in the article. The noise properties of the nonlinear optical frequency conversion are an important issue for the application to quantum network. Thus this should be discussed in the article.

3) In addition, pump power dependency of the conversion efficiency and the amount of noise should be provided because they are also important for the quantum properties of the converted light.

We added a clarification to the passage in the text on page 3. There is no degradation of the $g(2)(0)$: We measured the value before and after frequency conversion and found it to be unchanged.

Minor Comments:

1) The definitions of the following variables should be clearly described. η^{open}_t and η^{blocked}_t in eq. (2), η_{SNSPD} and η_{SiAPD} in eq. (3)

We clarified the definitions and added detector efficiencies needed. We added the labels used for the different modes in the setup to the setup sketch to provide more clarity.

2) 4-f spectral filter should be clearly described.

Both 4-f filter lines are described in the methods section on page 5.

Answer to reviewer #2:

The manuscript describes both frequency conversion and bandwidth compression of quantum light by a nonlinear process in a Titanium-diffused Lithium Niobate waveguide. Frequency and bandwidth manipulation are vital tools in the construction of a quantum network as they can link various components with disparate wavelength requirements.

The conversion is achieved with an overall efficiency greater than what could be attained with spectral filtering, which is ascertained by group velocity matching and choosing an appropriate pump bandwidth. The authors verify that the second-order correlation function of the output light from the conversion is well below the classical value of 1, and measure a spectrum of the output confirming bandwidth compression. The authors then discuss how to adapt their setup for use with a diamond NV memory.

The results are clear, novel, and of immediate interest to the quantum optics community. The paper is well-written and accessible to a broad audience. I recommend publication in Nature Communications after the authors address the following issues:

We thank the reviewer for the comments which help us to improve the manuscript. We are happy to provide further details on the experiment. We added key numbers to the main text, in addition we provide a methods section in the end of the paper going into detail regarding his/her comments.

1. The paper is lacking experimental detail. Most notably, there is no mention of photon detection rates. What is the coincident photon rate coming out of the PDC source? After the QPG? How many photon counts were required for the g_2 measurement?

We provide count numbers and count rates in the methods section on page 6.

Other missing experimental details include:

- a description of the 4-f spectral filter for the source pump
- the resolution of the time-of-flight spectrometer
- coincidence detection window used in the experiment

A small section describing both the 4f filters and the TOF spectrometer have been added to the methods section. The coincidence window is the same for all measurements and is given in the section on coincidence measurements.

2. The authors report that photon pairs from the source are “indeed spectrally decorrelated”. A numerical measure of separability of the initial JSI should be given to support this. Reporting the Schmidt number would suffice.

The Schmidt number of the state was extracted from the measurement data and added to the text on page 2.

3. When discussing bandwidth compression by chirped pulse upconversion, reference should also be given to PRA 91, 033809 (2015), which discusses how high efficiency can be achieved with this method.

Thank you for pointing out the reference, we added it at the appropriate position.

REVIEWERS' COMMENTS:

Reviewer #1 (Remarks to the Author):

The authors clearly addressed all the comments from the referees and revised the manuscript. The revised article is well written and the experimental details are clearly presented. I recommend the publication of this article in Nature Communications.

Reviewer #2 (Remarks to the Author):

The revised manuscript now includes sufficient experimental detail. As such, I recommend it for publication in Nature Communications.